

# Projected changes in the seasonal cycle of the Atlantic meridional heat transport in MPI-ESM

Matthias Fischer[1], Daniela I.V. Domeisen[2], Wolfgang A. Müller[3], and Johanna Baehr[1]

[1]Institute of Oceanography, Center for Earth System Research and Sustainability, University of Hamburg, Bundesstrasse 53, 20146 Hamburg, Germany.
[2]GEOMAR Helmholtz Centre for Ocean Research Kiel, Kiel, Germany.
[3]Max Planck Institute for Meteorology, Hamburg, Germany.

*Correspondence to:* Matthias Fischer (matthias.fischer@uni-hamburg.de)

**Abstract.** We investigate the effect of a projected reduction in the Atlantic Ocean meridional heat transport (OHT) on changes in its seasonal cycle. We analyze a climate projection experiment with the Max-Planck Institute Earth System Model (MPI-ESM) performed for the Coupled Model Intercomparison Project phase 5 (CMIP5). In the RCP8.5 climate change scenario, the OHT declines in MPI-ESM in the North Atlantic by 30-50% by the end of the 23rd century. The decline in the OHT is

accompanied by a change in the seasonal cycle of the total OHT and its components. We decompose the OHT into overturning and gyre component. For the total OHT seasonal cycle, we find a northward shift of 5 degrees and latitude dependent temporal shifts of 1 to 6 months that are mainly associated with changes in the meridional velocity field. We find that the shift in the OHT seasonal cycle predominantly results from changes in the wind-driven surface circulation which projects onto the overturning component of the OHT in the tropical and subtropical North Atlantic. This leads to latitude dependent shifts of 1 to 6 months

in the overturning component. In the subpolar North Atlantic, we find that the reduction of the North Atlantic Deep Water formation in RCP8.5 and changes in the gyre heat transport result in a strongly weakened seasonal cycle with a weakened seasonal amplitude by the end of the 23rd century and thus changes the OHT seasonal cycle in the SPG.

## 1 Introduction

Global surface temperatures are projected to warm - depending on the considered climate change scenario - intensively over

the next centuries (IPCC, 2013) accompanied by a projected shift in the amplitude and phase of the seasonal cycle of surface air temperatures (Dwyer et al., 2012). In concert, the Atlantic meridional overturning circulation (AMOC) is projected to slow down (Weaver et al., 2012; IPCC, 2013) which can be attributed to a reduction of deep water formation in the North Atlantic, especially in the Labrador Sea and Greenland Sea (Vellinga and Wood, 2002). The associated Atlantic Ocean meridional heat transport (OHT) is also thought to weaken due to the direct linear relation of AMOC and OHT found in observations and model

studies (Johns et al., 2011; Msadek et al., 2013). However, it is unclear how climate change along with a projected shift in the seasonal cycle of surface temperatures affects the seasonal cycle of the ocean circulation, and especially of the OHT. Here, we investigate projected changes in the OHT seasonal cycle in a Coupled Model Intercomparison Project phase 5 (CMIP5) climate projection (Taylor et al., 2012) performed in the global coupled Max-Planck Institute Earth System Model (MPI-ESM) .



In the CMIP5 Representative Concentration Pathway (RCP) RCP8.5, surface air temperatures are expected to increase by about 8 degrees in the global mean by the year 2300 in the CMIP5 multi-model ensemble (IPCC, 2013). The warming manifests itself over the continents and in particular in polar regions where an increase in surface temperatures of more than 20°C arises in climate projections until 2300 (e.g., IPCC, 2013; Bintanja and Van der Linden, 2013). Due to the strong warming in polar lati-

tudes the meridional temperature gradient from the equator to the poles is also strongly reduced in the Northern Hemisphere. The atmospheric circulation patterns are projected to move poleward in concert with the warming of surface temperature, leading to a poleward expansion of the tropical cell and a poleward shift of the jet stream and storm track (Chang et al., 2012; Hu et al., 2013; IPCC, 2013). A warmer planet has been shown to lead to an expansion of the Hadley cell and a poleward shift of the westerlies in both dynamical core (Butler et al., 2010) and complex climate models (Lu et al., 2008), following a systemat-

ically warmer Northern Hemisphere (e.g. Toggweiler, 2009). Under global warming, the hemispheric temperature asymmetry increases, leading to an additional northward shift of the ITCZ and the position of the westerlies. A number of mechanisms have been proposed for the shift of the Hadley circulation and the westerlies (Lu et al., 2014, and references therein).

In contrast to the general warming, the surface air temperatures show a prominent area of reduced warming over the North Atlantic subpolar gyre (SPG) in the set of CMIP5 climate projections that might be associated with an adjustment of the Atlantic

meridional overturning circulation (Drijfhout et al., 2012) or a reduction of the OHT into the SPG (e.g., Rahmstorf et al., 2015). These changes in the surface temperature patterns thus suggest considerable changes in the North Atlantic Ocean circulation, the AMOC and the associated OHT.

The implications of the Atlantic Ocean circulation and the OHT for the North Atlantic sector and the European climate have been widely discussed. The AMOC and OHT in the North Atlantic have been shown to affect the North Atlantic heat con-

tent and the North Atlantic sea surface temperatures (SST; e.g., Dong and Sutton, 2003; Grist et al., 2010; Sonnewald et al., 2013; Muir and Fedorov, 2014). Changes in the North Atlantic SSTs and the air-sea interaction appear to be important for influencing the atmospheric circulation, the multidecadal variability of the North Atlantic sector and the North American and European climate on inter-annual to multi-decadal time scales (e.g. Rodwell et al., 2004; Sutton and Hodson, 2005; Gastineau and Frankignoul, 2015).

Further, a response of the NAO to North Atlantic sea surface temperatures has been found both in observations and model studies (Czaja et al., 1999; Czaja and Frankignoul, 2002; Rodwell and Folland, 2002; Frankignoul et al., 2013; Gastineau et al., 2013; Gastineau and Frankignoul, 2015). Via the Atlantic Multidecadal Oscillation (AMO), which is thought to be associated with AMOC and OHT variability (e.g., Delworth and Greatbatch, 2000; Knight et al., 2005; Msadek and Frankignoul, 2009; Zhang and Wang, 2013), the SST variability has been linked to a number of climate phenomena, such as Sahel rainfall, Atlantic

hurricane activity and North American and European summer climate (Enfield et al., 2001; Sutton and Hodson, 2005; Knight et al., 2006; Zhang and Delworth, 2006; Sutton and Dong, 2012). Recently, Clement et al. (2015) reported that the AMO can be reproduced in an atmospheric circulation model coupled to a slab ocean without changes in the ocean circulation and heat transport. They showed in their model that the AMO is a response to the atmospheric circulation in the mid-latitudes rather than to the ocean. However, the specific role and direct importance of the OHT for European climate is still controversially

discussed and the exact mechanism not fully understood (e.g., Bryden, 1993; Seager et al., 2002; Rhines et al., 2008; Riser and





Lozier, 2013). The seasonal coupling between ocean and atmosphere is less understood. Minobe et al. (2010) have shown an atmospheric response to Gulf Stream variability with seasonal variations. When considering also the impact of seasonal variations in the total OHT on European climate, the relation becomes even more complex and thus requires a better understanding of the OHT and its coupling to the atmosphere.

Most of the present understanding stems from model analysis, due to a lack of continuous observations. These observations of the OHT rely on hydrographic snapshots (e.g., Bryan, 1962; Hall and Bryden, 1982; Lavin et al., 1998; Lumpkin and Speer, 2007) or inverse methods (e.g., Macdonald and Wunsch, 1996; Ganachaud and Wunsch, 2000, 2003) and give estimates of the time mean OHT of about 1 PW at its maximum at about 20°N, but do not describe the OHT variability (see also Wunsch, 2005). Further, single hydrographic snapshots may be affected by a seasonal bias due to the predominance of field work during

summer. Recently, the two time series of the 26°N Rapid array and observations at 41°N have indicated long-term variability and a clear seasonal cycle of the OHT in the North Atlantic (Johns et al., 2011; Hobbs and Willis, 2012).

Model studies led to a better understanding of the dynamics of the seasonal cycle of the OHT. The pioneering study by Bryan (1982a) used a global ocean circulation model forced with observed winds. Bryan pointed out the importance of the wind-driven Ekman mass transport and of the associated Ekman heat transport for driving the seasonal variability of the OHT, which

was also found in subsequent studies (Sarmiento, 1986; Lee and Marotzke, 1998; Jayne and Marotzke, 2001; Böning et al., 2001; Cabanes et al., 2008; Balan Sarojini et al., 2011; Munoz et al., 2011). Bryan argued that changes in the zonally integrated wind stress, leading to changes in the Ekman mass transport, are balanced by a barotropic return flow. Jayne and Marotzke (2001) provided the theoretical and dynamical justification for Bryan's argumentation, stressing again the important role of the Ekman transport for the seasonal cycle of the OHT.

Traditionally, the OHT is decomposed into a vertical overturning component, which is commonly linked to the large scale overturning, and a horizontal gyre component giving correlations of the zonal deviations of the velocity and temperature field (Bryan, 1962, 1982b; Bryden and Imawaki, 2001; Siedler et al., 2013). The gyre component is commonly linked to the horizontal gyre circulation and contributions from the eddy field. Previous studies have shown that the overturning component dominates the time mean, as well as the interdecadal variability of the OHT in the tropical and subtropical North Atlantic,

whereas the overturning and gyre component contribute about equally to the OHT and its interdecadal variability in the subpolar North Atlantic (e.g., Eden and Jung, 2001).

With this study, we aim to understand how the seasonal cycle of the Atlantic Ocean meridional heat transport is affected by global warming and what determines potential changes in the OHT seasonal cycle. For our analysis, we use a CMIP5 climate change projection performed in MPI-ESM, with a focus on the climate change scenario RCP8.5. We aim to identify changes

in the seasonal cycle of OHT and its sources. To analyze different physical mechanisms that contribute to the changes in the seasonal cycle, we analyze the individual contributions to the total OHT on seasonal time scales. Therefore, we decompose the OHT into gyre- and overturning component, related to the horizontal gyre circulation and to the overturning circulation in the North Atlantic and consider changes in the wind-driven Ekman heat transport.

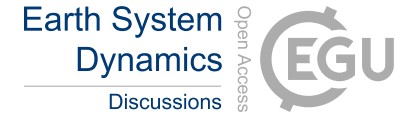

## 2    Model and Methods

### 2.1    The CMIP5 climate change scenario RCP8.5 in MPI-ESM

We analyze climate projection experiments of the CMIP5 ensemble (Taylor et al., 2012) performed in the coupled Max-Planck-Institute Earth System model in low resolution configuration (MPI-ESM-LR) integrated from 1850 to 2300. MPI-ESM-LR
comprises MPIOM for the ocean component and ECHAM6 for the atmospheric component (Marsland et al., 2003; Jungclaus et al., 2013; Stevens et al., 2013). In MPIOM, the horizontal resolution is 1.5 degree on average with 40 unevenly spaced vertical levels (Marsland et al., 2003; Jungclaus et al., 2013). ECHAM6 has a horizontal resolution of T63 and includes 47 vertical levels (Stevens et al., 2013).

For our analysis, we focus on one member in the CMIP5 ensemble and use the historical simulation (1850-2005) extended
with the Representative Concentration Pathway RCP8.5 from 2006 to 2300. In RCP8.5, a rising radiative forcing following "business as usual" is applied, which rises to $8.5 W/m^2$ in the year 2100, and further increases after that (van Vuuren et al., 2011). We focus in this study on long term changes in RCP8.5, comparing the period 1850-1950 for the historical simulation ($\text{HIST}_{mean}$) to the period 2200-2300 for the RCP8.5 scenario ($\text{RCP}_{mean}$), where we expect the strongest changes in the North Atlantic Ocean circulation and in the seasonal cycle of the OHT.

### 2.1.1    Projected changes in the North Atlantic sea surface temperatures

In concert with the projected warming of surface air temperatures, the sea surface temperatures (SST) are projected to warm globally and also in the North Atlantic sector in RCP8.5 (Fig.1). A similar "warming hole" signature as found for surface air temperatures (c.f., Drijfhout et al., 2012) is present in the North Atlantic SSTs (Fig.1) with a stronger warming in polar regions and an area of reduced warming in the SPG (Fig.1c). Pronounced regional variations of the SST change suggest important
changes in the North Atlantic Ocean circulation and its dynamics. The SST front along the Gulf Stream/North Atlantic current path shifts northward and weakens which might also impact the North Atlantic storm track as already shown for the current climate state (e.g. Minobe et al., 2008, 2010; Hand et al., 2014).

### 2.1.2    Projected changes in the North Atlantic horizontal gyre circulation and zonal-mean zonal wind

The area of reduced warming in the eastern SPG indicates changes in the North Atlantic Ocean dynamics and in the gyre
circulation (e.g., Drijfhout et al., 2012). The North Atlantic barotropic stream function shows substantial changes in the annual mean pattern (Fig.2). The barotropic stream function weakens in the subtropical gyre and intensifies in the SPG in RCP8.5. We identify a northward shift of the subtropical gyre and a northward shift of the boundary between subtropical and subpolar gyre by about 5 degrees between the $\text{HIST}_{mean}$ (Fig.2a) and $\text{RCP}_{mean}$ (Fig.2b) associated with the northward shift of the atmospheric wind field (Fig.2c).

The zonal-mean zonal wind across the Atlantic indicates considerable changes in the annual mean surface wind field in RCP8.5 (Fig.2c,3). As compared to the $\text{HIST}_{mean}$, the northern Hadley cell slightly expands poleward and equatorward and the Ferrel





cell shifts poleward in RCP$_{mean}$ in MPI-ESM (Fig.3a) as in most CMIP5 models (e.g., Hu et al., 2013). As a consequence, the westerlies between 30°N and 60°N are shifted poleward in RCP8.5 by about 5 degrees (Fig.3b,c). This shift resembles a positive NAO anomaly, which is associated with an acceleration of the westerlies over large areas of the SPG (Fig.3b,c), along with a deceleration of the westerlies between 30°N - 40°N and a slight intensification of the trade winds south of 30°N.

In concert with this intensification of the surface wind field the circulation of the SPG strengthens with an increase of the average transport by about 2 Sv, which might be related to changes in heat fluxes in the SPG (e.g. Eden and Willebrand, 2001; Eden and Jung, 2001; Barrier et al., 2014). In particular, the flat-bottom Sverdrup transport in the subpolar gyre indicates only a weak increase of about 0.5 Sv in the gyre strength from HIST$_{mean}$ to RCP$_{mean}$ (not shown), suggesting that changes in the deep circulation are important (Greatbatch et al., 1991). The subtropical gyre shows a weakening in the barotropic

streamfunction by about 20 Sv at its maximum at about 30°N and by about 4 Sv in its mean indicating important changes in the dynamics of the subtropical gyre (Fig.2). Considering the Sverdrup transport in the subtropical gyre, we find a decrease in the mean by about 1.5 Sv, while the maximum is reduced by roughly 10 Sv. In concert with the northward shift of the ocean circulation in RCP8.5 the North Atlantic current moves further north in RCP8.5. This leads to the simulated changes in the SST front (Fig.1).

## 2.2 The Atlantic meridional heat transport and its decomposition

Traditionally, the meridional heat transport **H** is diagnosed from the zonal and vertical integral of the heat flux across an east-west section through the Atlantic (e.g., Hall and Bryden, 1982):

$$\mathbf{H}(y) = \rho_0 c_p \int_{x_W}^{x_E} \int_{-H(x,y)}^{0} v(x,y,z) \theta(x,y,z) \, dz \, dx, \tag{1}$$

with $\rho_0$ a reference density, $c_p$ the specific heat capacity of sea water, $H$ the water depth, $x$ the longitude, $y$ the latitude, $z$ the
depth, $x_E$ and $x_W$ the eastern and western boundaries of the transect, $v$ the meridional velocity and $\theta$ the potential temperature in °C.

### 2.2.1 Impact of the variability of the temperature and velocity field on the OHT

In order to assess the impact of temporal variations in the velocity field and in the potential temperature field, we separate their contributions to the OHT. In a first step we calculate the OHT with a time mean velocity field ($[v]$, Eq. 2), and in a second
step with a time mean temperature field ($[\theta]$, Eq. 3) over the analyzed periods HIST$_{mean}$ and RCP$_{mean}$. We consider the time mean of the $v$- ($\theta$-) field but consider the full spatial variations of the respective field together with the full spatial and temporal





variability of the $\theta$- (v-) field, such that the two contributions can be calculated from

$$\mathbf{H}_{[v]}(y) = \rho_0 \, c_p \int\limits_{x_W}^{x_E} \int\limits_{-H(x,y,z)}^{0} [v(x,y,z)] \, \theta(x,y,z,t) \, dz \, dx \qquad (2)$$

$$\mathbf{H}_{[\theta]}(y) = \rho_0 \, c_p \int\limits_{x_W}^{x_E} \int\limits_{-H(x,y,z)}^{0} v(x,y,z,t) \, [\theta(x,y,z)] \, dz \, dx \qquad (3)$$

with $v$ the meridional velocity, $\theta$ the temperature and $[v]$ and $[\theta]$ the time mean of the velocity and temperature (°C) field

5  over the analyzed periods $\text{HIST}_{mean}$ and $\text{RCP}_{mean}$. The two cases correspond to the time mean velocity field advecting the time-dependent temperature field and the time-dependent velocity field acting on the time mean temperature field. Based on this split-up of the OHT we then analyze the impact of the variability in the velocity and temperature field on the seasonal cycle of the OHT.

### 2.2.2 Overturning and gyre component of the OHT

10  Well-established is the decomposition of the OHT into contributions from the zonal mean vertical circulation and the horizontal circulation by considering the zonal mean $(\overline{v}, \overline{\theta})$ and deviations from the zonal mean $(v', \theta')$ of the meridional velocity and temperature field respectively: $v = \overline{v} + v'$ and $\theta = \overline{\theta} + \theta'$ (e.g., Bryan, 1962, 1982b; Bryden and Imawaki, 2001). This yields for the OHT

$$
\mathbf{H}(y) = \quad \rho_0 c_p \underbrace{\int\limits_{x_W}^{x_E} \int\limits_{-H(x,y)}^{0} \overline{v(x,y,z)} \; \overline{\theta(x,y,z)} \, dz \, dx}_{\mathbf{H}^{ov} \, = \, \text{overturning component}}
$$

$$
+ \quad \rho_0 c_p \underbrace{\int\limits_{x_W}^{x_E} \int\limits_{-H(x,y)}^{0} v'(x,y,z)\theta'(x,y,z) \, dz \, dx}_{\mathbf{H}^{gyre} \, = \, \text{gyre component}} \qquad (4)
$$

giving an overturning component $\mathbf{H}^{ov}$ and a gyre component $\mathbf{H}^{gyre}$ from the horizontal gyre circulation. As the total OHT, both components hold mass balance by definition for a closed basin. Traditionally, the overturning component is related to the zonally averaged vertical-meridional (overturning) circulation and the gyre component is related to the horizontal transport by the large-scale gyres and small-scale eddies.

20  Further, an Ekman heat transport contribution to the overturning heat transport can be calculated from

$$\mathbf{H}_{ek}^{ov}(y) = -c_p \int\limits_{x_W}^{x_E} \frac{\tau_x(x,y)}{f(y)} \left( \theta_{ek}(x,y) - \langle \theta(x,y,z) \rangle \right) dx, \qquad (5)$$

with $\tau_x$ the zonal wind stress, $f$ the Coriolis parameter, $\langle \theta \rangle$ the temperature field averaged zonally and vertically across the section and $\theta_{ek}$ the temperature of the Ekman layer following Böning and Hermann (1994). Here, the Ekman heat transport at




the surface is assumed to be compensated by a deep return flow. We also assume $\theta_{ek}$ to be close to the surface temperature, which yields only small uncertainties (Johns et al., 2011). Williams et al. (2014) analyzed contributions from the overturning, gyre and Ekman heat transport to the heat convergence in the North Atlantic for decadal signals based on perturbation experiments with and without wind. Thus, they avoid the assumption of a uniform return flow as done in Eq.5. Jayne and Marotzke

(2001) showed the computation of the Ekman heat transport conserves mass only for short time scales of some weeks, but not necessarily for the time mean heat transport, so that we apply the Ekman transport calculation only to the OHT seasonal variability and not to the time mean OHT.

## 3 Mean changes in the Atlantic meridional overturning circulation and meridional heat transport

### 3.1 AMOC

The mean changes seen in the SSTs, the surface wind field and in the North Atlantic Ocean circulation influence the AMOC and the OHT, which we focus on in the remainder of the study. The AMOC shows significant changes in the time mean from $\text{HIST}_{mean}$ to $\text{RCP}_{mean}$ (Fig.4). The AMOC calculated in depth coordinates shows that the northward overturning cell is reduced and shifted to the surface from the $\text{HIST}_{mean}$ to $\text{RCP}_{mean}$ (Fig.4a,b). The maximum $\psi_{max}$ of the stream function $\psi(y,z) = \int_{z}^{0} \int_{x_W}^{x_E} v(x,y,z)\, dx\, dz$ commonly used as an index for the AMOC, is substantially reduced between 30% and 50% in

the North Atlantic from $\text{HIST}_{mean}$ to $\text{RCP}_{mean}$ (Fig.4a,b;Fig.5a).

Considering the AMOC in density coordinates (Fig.4c,d) indicates a similar surfaceward shift of the AMOC cell to layers of lower density from $\text{HIST}_{mean}$ to $\text{RCP}_{mean}$ (Fig.4c,d). We find only a slight decrease of the wind-driven surface cell in the tropics by about $2\ Sv$ at the maximum, whereas the deep cell is reduced by more than 50% from a maximum of about $24\ Sv$ in $\text{HIST}_{mean}$ to about $10\ Sv$ in $\text{RCP}_{mean}$. In RCP8.5, the formation of NADW in the Labrador Sea and the Nordic seas is

almost absent for the 2200-2300 period. Instead of deep convection mixing surface water down to the bottom (about 3000m depth in the Labrador Basin and Irminger Basin) in the historical simulation, the maximum mixed layer depth is mostly limited to the upper 1000 meters in RCP8.5 (not shown), which thus directly reduces the deep branch of the AMOC. In addition, the AMOC's weakening is associated with a reduction of the geostrophic volume transport (Fig.5a). For simplicity, we approximate the maximum geostrophic transport $\psi_{geo}$ by the residual of $\psi_{max}$ and the Ekman transport $\psi_{ek}$ given by $\psi_{ek} = -\frac{1}{\rho_0 f} \int_{x_W}^{x_E} \tau_x\, dx$

with $\tau_x$ the zonal wind stress at the ocean surface: $\psi_{geo} \approx \psi_{max} - \psi_{ek}$. The geostrophic transport is proportional to the zonal cross-basin density gradient which is decreased from $\text{HIST}_{mean}$ to $\text{RCP}_{mean}$ and thus reduces the AMOC in the North Atlantic (not shown). The Ekman transport indicates only small and local changes from $\text{HIST}_{mean}$ to $\text{RCP}_{mean}$ that do not contribute significantly to the weakening of the AMOC (Fig.5a).

### 3.2 OHT

Similar to the AMOC, the RCP8.5 scenario reveals considerable changes in the associated OHT. For $\text{RCP}_{mean}$, the OHT shows a pronounced weakening by 30-50% from about 1.2 PW to about 0.8 PW between 10°N and 30°N and from about 0.8



PW to about 0.4PW between 40°N and 55°N by the end of the 23rd century (Fig.5b). The reduction in the total OHT in the subtropical North Atlantic can be attributed almost entirely to a reduction in the overturning heat transport, while changes in the gyre component are comparably small. Only in the SPG, the gyre component also indicates a substantial weakening, so that both the overturning and the gyre component contribute to the reduction in the total heat transport in the subpolar North

Atlantic. The reduction of the overturning heat transport can be attributed to a reduction of the geostrophic contribution to the AMOC (Fig.5a) and the associated reduction of the zonally-averaged geostrophic meridional velocity field.

## 4  Changes in the seasonal cycle of the Atlantic meridional heat transport

### 4.1  The total OHT

To assess the response of the seasonal cycle of the OHT to a changing climate in RCP8.5, we first analyze the latitude dependent

seasonal cycle of the total OHT before focusing on the seasonal cycle of individual OHT components. The seasonal cycle of the OHT shows regionally varying patterns with a seasonal amplitude declining from the equator the pole and phase changes between the tropical, subtropical and subpolar North Atlantic (Fig.6). The most obvious change in the OHT from the $HIST_{mean}$ to $RCP_{mean}$ is the reduction of the mean heat transport, which appears in almost all months (Fig.6a,b). Since the changed seasonal cycle is superimposed on the strong reduction of the OHT, we consider in the following analysis anomalies of the

seasonal cycle relative to the annual mean at every latitude (Fig.6c,d).

The seasonal anomalies indicate changes in space and time in the OHT seasonal cycle from the $HIST_{mean}$ to $RCP_{mean}$ (Fig.6c,d). The OHT seasonal cycle pattern shows a northward shift by about 5 degrees following the general northward shift of the atmospheric jet and the gyre circulation in RCP8.5. We also find a latitude dependent temporal shift of 1 to 6 months of the minima and maxima of the seasonal cycle that can not be fully explained by the northward shift of the pattern. The

temporal shift appears to be different between the tropical, subtropical and subpolar North Atlantic. Especially latitudes along the gyre boundaries between the tropical and subtropical North Atlantic (at about 20°N) and the subtropical and subpolar North Atlantic (at about 40°N) indicate significant phase shifts of 4 to 6 months that mostly result from the northward shift here.

In addition, we find changes in the seasonal amplitude in $RCP_{mean}$ which also depend on latitude and are partly influenced by the northward shift. Between 30°N-40°N, the seasonal cycle generally exhibits an intensification in the amplitude, whereas

the seasonal amplitude between 40°N-50°N is influenced mostly by the northward shift. As an example for the subtropical and subpolar gyre, the OHT seasonal cycle is shown at 30°N and 45°N from the $HIST_{mean}$ to $RCP_{mean}$ (Fig.6e-f) showing prominent changes in the amplitude, the phase and the general seasonality of the OHT.

### 4.1.1  Contributions from the seasonal variability in the temperature and velocity field

To identify whether changes in the seasonal cycle of the velocity field or in the temperature field dominate the changes seen in

the total OHT, we consider the OHT with a time-mean velocity field $[v]$ (Eq.2) allowing for temporal –also seasonal– variability in the potential temperature field and a time-mean temperature field $[\theta]$ allowing for temporal variability in the velocity field



(Eq.3), so that the non-time-mean component provides the seasonal variability only. The OHT based on $[v]$ (Fig.7a,b) reveals a reduced seasonality compared to the full OHT seasonal variability, especially in the tropical and subtropical North Atlantic. The changes in the seasonal cycle from $\text{HIST}_{mean}$ to $\text{RCP}_{mean}$ are rather small. The OHT based on $[\theta]$ (Fig.7c,d) reproduces the bulk of the total OHT seasonal cycle and also the changes in the seasonal cycle from $\text{HIST}_{mean}$ to $\text{RCP}_{mean}$. This clearly

indicates that the strongest changes in the OHT seasonal cycle mostly result from changes in the meridional velocity field, whereas the overall warming of the ocean temperatures plays a less important role in directly changing the OHT seasonal cycle via the temperature field.

### 4.1.2 Zonal-mean zonal wind and Ekman heat transport

The seasonal cycle of the zonal-mean zonal wind indicates a seasonal maximum of the atmospheric westerly jet in winter

and meridional shifts of the position of the jet from summer to winter in $\text{HIST}_{mean}$ (Fig.8; shown is the full zonal-mean zonal velocity field). Especially in the tropical Atlantic, the seasonality of the wind field is strongly affected by the seasonal migration of the ITCZ (e.g., Schneider et al., 2014). Between $\text{HIST}_{mean}$ and $\text{RCP}_{mean}$ the zonal wind undergoes changes in amplitude and position of the jet with associated temporal changes in the seasonal cycle (Fig.8a,b; c.f. Lu et al., 2014). We find a seasonally dependent shift and expansion of the Hadley cell and a northward shift of the Ferrel cell. During winter, the

westerlies are shifted northward by about 5 degrees from $\text{HIST}_{mean}$ to $\text{RCP}_{mean}$. In contrast to the changes in winter, we find a general broadening of the westerlies during summer in $\text{RCP}_{mean}$, corresponding to a southward shift of the trade wind regime by about 2 degrees and a poleward shift of the maximum westerlies for $\text{RCP}_{mean}$ (Fig.8a,b). Changes of the zonal wind during summer lead to reduced easterly winds over the subtropical gyre, reduced westerlies between 40°N and 50°N and enhanced westerlies north of 50°N during summer (Fig.8a,b).

The seasonal cycle of the Ekman heat transport indicates a weakening in the seasonal cycle in the tropical North Atlantic with a decrease in the seasonal amplitude by about 50% from $\text{HIST}_{mean}$ to $\text{RCP}_{mean}$ (Fig.8e-h). In the subtropical gyre, we find a dominant influence of the northward shifted westerlies on the Ekman heat transport. The Ekman heat transport in the SPG shows – in contrast to the subtropical gyre – relatively small changes in terms of the amplitude resulting in a slight strengthening in summer and a weakening in winter (Fig.8e,f). As an example, the Ekman heat transport seasonal cycle is shown at 30°N and

45°N (Fig.8g,h) indicating the influence of the northward shifted pattern.

The changes in the seasonal amplitude of the Ekman heat transport come in concert with a temporal shift of the seasonal minima and maxima (Fig.8e-h). The Ekman heat transport in the tropical North Atlantic undergoes a 1-2 months temporal shift to later months. In the southern part of the subtropical gyre (about 20°N-30°N), we find the largest temporal shift of the seasonal maximum and minimum of 2-6 months to later months (Fig.8e). In the northern part, the maximum is shifted by 1-2

months, as is the minimum. The subpolar gyre region shows only small changes in the Ekman heat transport seasonal cycle (1-2 months), while a latitude-dependent larger shift of about 5 months is identified for the maximum at about 40°N due to the northward shift of the pattern along the gyre boundary (e.g., Fig.8f). The shift considerably changes the seasonal cycle of the Ekman heat transport depending on latitude, closely following the seasonal cycle of the surface wind.




### 4.1.3 Overturning and gyre heat transport

The overturning and gyre component show that similar to the time mean and long term variability the overturning component dominates the OHT seasonal cycle in the subtropical North Atlantic Fig.8a,b), while the gyre component gains influence in the subpolar gyre (Fig.8c,d). The changes in the seasonal cycle of the overturning component from $HIST_{mean}$ to $RCP_{mean}$

therefore reveal clear similarities to the changes in the seasonal cycle of the total OHT (Fig.6). We find a similar northward shift of the seasonal cycle pattern by about 5 degrees - suggesting a relation to the surface wind field - and comparable changes to the OHT in the seasonal amplitude with a 2-4 months shift of the minimum and maximum in the subtropical gyre and up to 6 months shift in the subpolar gyre. This close relation shows that changes in the seasonal cycle of the overturning component drive the changes in the seasonal cycle of the total OHT in both the subtropical and subpolar gyre (Fig.8a,b). Similarly, the

overturning component determines changes in the seasonal amplitude of the total OHT, with a reduction in the seasonal amplitude in the tropics and a slight increase of the amplitude between $30°N$ and $45°N$.

In $RCP_{mean}$ (Fig.8c,d), the gyre component reveals a slight intensification of the seasonal amplitude in tropical latitudes, while no significant changes in the seasonal amplitude occur in the subtropical and subpolar gyre. Important changes for the gyre component's seasonal cycle take place at about $40°N$ where the gyre boundary is situated in the model. We find a northward

shift in the seasonal cycle pattern in the subpolar gyre following the northward shift in the barotropic stream function and the zonal-mean zonal wind (Fig.2) with the seasonal cycle in the subpolar gyre covering latitudes north of $40°N$ in $HIST_{mean}$ while the seasonal cycle covering latitudes north of $45°N$ in $RCP_{mean}$ (Fig.8c,d).

The comparison of the changes in the OHT, the overturning component (Fig.8a,b) and the Ekman heat transport reveals that changes in the Ekman heat transport (Fig.8e,f) can explain a large part of the changes in the seasonal cycle of the OHT and

overturning component: on the one hand by the Ekman heat transport's seasonal cycle contributing to the overturning component, on the other hand earlier studies have shown effects from wind stress on the vertical motion (heaving and shoaling) of isopycnals (Köhl, 2005; Chidichimo et al., 2010; Kanzow et al., 2010). Thereby, the surface wind stress might change the interior geostrophic flow and hence the heat transport and its variability. Overall, changes in the seasonal cycle are predominantly driven by changes in the ocean's surface and upper ocean, as also found in the seasonal cycle of the temperature transport in

potential density coordinates (appendix A), indicating changes in the surface and intermediate circulation.

## 5   Discussion

The changes in the mean climate state of the North Atlantic and a projected reduction in the AMOC and OHT in MPI-ESM come in concert with changes in the seasonal cycle of the OHT. Bryan (1982a) and subsequent studies have shown that the Ekman (heat) transport is responsible for a large fraction of the seasonal variability of the overturning heat transport and thus

of the total oceanic OHT. We have shown that under climate change the overturning heat transport constitutes the prominent factor for the OHT seasonal cycle on the one hand, and that the overturning heat transport is also the most important term leading to the changes in the OHT seasonal cycle on the other hand. These changes in the overturning heat transport are mostly wind-driven by the Ekman heat transport mostly confined to the upper layers of the ocean and might also be associated with



changes in the geostrophic interior flow from a wind-driven heaving and shoaling of the isopycnal slope, as shown for the AMOC seasonal cycle in observations (Kanzow et al., 2010), as well as changes in the water mass characteristics (appendix A). Changes in the Ekman transport and the associated vertical Ekman velocities change the isopycnal slope and thus the geostrophic velocity field. Overall the seasonal cycle of the OHT largely adjusts to a changed seasonality of the atmospheric

circulation and the zonal wind in RCP8.5. Similar changes in the seasonal cycle for extreme climate change scenarios have also been found in other atmospheric variables such as surface temperatures and precipitation (Dwyer et al., 2012; Donohoe and Battisti, 2013; Dwyer et al., 2014).

Most prominent among the atmospheric changes with climate change is the expansion of the Hadley cell and the associated northward shift of the ITCZ and the mid-latitude westerlies (Sun et al., 2013; Lu et al., 2014). But the exact mechanism leading

to the shift of the ITCZ and the westerlies is still not fully understood and under discussion (Seidel et al., 2008), especially in CMIP5 models where the problem of a double ITCZ occurs in some models (Hwang and Frierson, 2013; Christensen et al., 2013). As shown by Hu et al. (2013) almost all CMIP5 models show a trend of poleward expansion of the Hadley cell in the RCP4.5 and RCP8.5 scenarios for the period 2006 to 2100. Hu et al. also show that the CMIP5 historical simulations underestimate the trend in the poleward expansion of the Hadley cell represented by reanalysis data for the preceding decades,

although it is unclear whether the trend is anthropogenically forced or whether the models capture the natural variability and extent of the Hadley cell correctly.

But changes in the surface winds and wind stress may be model dependent and may differ in detail, i.e. some models do not project a northward shift of the westerlies directly at the surface and in the associated surface wind stress. Thus, the proposed mechanism for changes in the seasonal cycle of the oceanic OHT by the Ekman heat transport and the associated changes in

the geostrophic velocity field might differ between individual models used for the CMIP5 multi-model ensemble and might require a similar analysis in other CMIP5 models.

The strong decrease of the mean overturning heat transport leading to the 30-50% decrease in the OHT suggests that either the reduced meridional temperature gradient requires less heat to be transported to the poles or that a compensation mechanism must be at work, bringing additional heat from the equator to the poles to obtain a closed heat budget. In MPI-ESM, the

atmosphere compensates the decrease in the meridional ocean heat transport, implying an increased atmospheric heat transport (not shown), as also suggested by Rose and Ferreira (2012). A deeper analysis of the atmospheric compensation and changes in the atmospheric heat transport is needed, but is beyond the scope of our study.

The advection of heat by the ocean determines ocean heat storage rates and is an important factor for air-sea heat exchange (Dong et al., 2007), and thus for carrying heat to the North Atlantic sector and especially towards the European continent. By

the changed ocean and heat transport dynamics, the surface air-sea heat fluxes are presumably exposed to changes regarding areas of heat flux divergences and convergences and thus of heat exchange and also shifts in the seasonal cycle of surface heat fluxes, which might affect the climate over Europe.

In agreement with other studies (e.g., Gregory et al., 2005), the cooling associated with the decline of the OHT and the AMOC is smaller than the radiative heating of the atmospheric temperatures due to global warming. This yields an overall increase in

surface temperature in the North Atlantic sector. Hence, it is difficult to clearly separate the effect of the reduced ocean heat





transport on surface temperatures from the increased radiative heating of surface temperatures. To identify this impact of the reduced OHT and changes in the OHT seasonal cycle, further studies will be required for clarifying the impact of a reduction and a changed seasonal cycle of the OHT on the North Atlantic sector and European climate.

## 6 Conclusions

Based on our analysis in the MPI-ESM CMIP5 climate projection RCP8.5, we conclude for the Atlantic Ocean meridional heat transport:

1. Along with a 30 to 50% decline of the time-mean OHT, the seasonal cycle of the OHT is shifted in time (1 to 6 months, depending on latitude and season) and in space (5° northward) in both the subtropical and subpolar gyre in RCP8.5.

2. These changes stem from a latitude-dependent altered seasonal cycle and a northward shift in the zonal-mean zonal wind (about 5° northward) and the resulting changes in the surface wind field that lead to a shift by 1 to 5 months in the seasonal cycle of the Ekman heat transport and the overturning heat transport.

3. Especially in the tropical and subtropical North Atlantic, the OHT seasonal cycle is mostly forced and mostly changed in the surface and intermediate layer, where the wind acts as the dominant direct driver of the seasonal variability and leads to temporal shifts from 1 to 6 months.

4. Thus, the changes in the total OHT seasonal cycle in the subtropical gyre result mostly from the wind-driven and surface-intensified part of the overturning heat transport, whereas in the subpolar gyre, the changes in the seasonal cycle are dominated by the gyre heat transport.

These findings may have important implications for the impact of climate change on the decadal predictability of the AMOC and the OHT.

## Appendix A: The meridional temperature transport in potential density coordinates

### A1 Methods

The decomposition of the OHT into overturning and gyre component merely represents the vertical integral and thereby masks out any contribution from different layers and water masses in the North Atlantic. To analyze how the vertical structure of the North Atlantic ocean circulation and associated changes in the water mass characteristics contribute to changes in the seasonal cycle of the OHT, we calculate the OHT in potential density coordinates, similar to the analysis of Talley (2003). Specifically, we calculate the temperature transport for chosen potential density ranges, since we can not ensure mass balance for every considered density class. The temperature transport $T$ in $PWT$ (1 $PWT = 1 \times 10^{15} W$) per density class is calculated from:

$$T(y, (\sigma_{2_i}, \sigma_{2_{i+1}})) = \rho_0 c_p \int_{x_W}^{x_E} \int_{z(x,y,\sigma_{2_i})}^{z(x,y,\sigma_{2_{i+1}})} v(x,y,z)\theta(x,y,z)dxdz \tag{A1}$$





with $\sigma_2$ the potential density referenced to 200 $dbar$, $v$ the meridional velocity, $\theta$ the potential temperature in $°C$. For every density class, the temperature transport is integrated between the depth of the upper and lower limit of that density class given by the depth of the respective isopycnal $z(x, y, \sigma_{2_i})$ and $z(x, y, \sigma_{2_{i+1}})$. For the temperature transport, the unit PWT is used to make clear the difference of the temperature transport to the mass balanced OHT. Even though the temperature transport

does not hold mass-balance, it is an appropriate choice for the calculation of the heat flux associated with the individual water masses. But for the full integral which is the sum of the individual components of $T$ and gives the OHT, mass is conserved. In contrast to Talley (2003), we use $\sigma_2$ as density.

Through the relation of the density, in particular of the zonal density gradient, to the geostrophic transport of the AMOC by the thermal wind relation we expect to find changes in the vertical structure where water mass properties and the potential

density changes. For the definition of individual water masses, we therefore perform a regression analysis for eastern boundary fields, western boundary fields and the zonal mean fields of $\theta$, $S$ and $\sigma_2$ on the AMOC at 26°N for $\text{HIST}_{mean}$ and $\text{RCP}_{mean}$ individually for annual mean values of $\theta$, $S$ and $\sigma_2$. The regression analysis then enables us to identify main water masses based on changes in the vertical profiles of the regression profile of $\theta$, $S$ and $\sigma_2$ on the AMOC (not shown) following Baehr et al. (2007).

Based on the regression analysis, we subdivide the temperature transport into four layers with fixed potential density ranges with water masses associated with the surface circulation, an intermediate layer, North Atlantic Deep Water (NADW, including parts of the lower Labrador Sea Water=LSW, Denmark Strait Overflow Water=DSOW and Iceland-Scotland Overflow Water=ISOW) and abyssal waters from the Antarctic Bottom Water (AABW) (see table 1). The temperature-salinity diagrams

reveal changes in the water mass properties from $\text{HIST}_{mean}$ to $\text{RCP}_{mean}$ with warmer and saltier waters for surface and intermediate layers in $\text{RCP}_{mean}$ than in $\text{HIST}_{mean}$ yielding layers of lighter density in $\text{RCP}_{mean}$ (Fig.10). Since we find changes in the density classes and the associated water mass characteristics between $\text{HIST}_{mean}$ and $\text{RCP}_{mean}$, the water mass definitions differ between the $\text{HIST}_{mean}$ and $\text{RCP}_{mean}$ and the individual water masses are therefore determined separately. In RCP8.5, the deep water formation in the North Atlantic is significantly reduced, leading to a change in the water mass distribution. It is

not convenient anymore to define a traditional North Atlantic Deep Water, which is why the density classes used to define the individual water masses differ between $\text{HIST}_{mean}$ and $\text{RCP}_{mean}$. A finer separation of individual water masses is not feasible in the model. For each water mass with the respective density range, we then calculate the temperature transport following Eq. A1 and the corresponding seasonal cycles.

The temperature transport for the individual water masses confirms that the northward heat transport is mostly confined to the

surface layer in the tropical and subtropical North Atlantic in $\text{HIST}_{mean}$ and $\text{RCP}_{mean}$ (Fig.11). The intermediate water temperature transport increases from the subtropical to the subpolar gyre and dominates the total OHT between 40°N and 55°N in $\text{HIST}_{mean}$ and between 40°N and 70°N in $\text{RCP}_{mean}$, reflecting the outcropping of the intermediate layer around 45°N. The NADW contributes with a southward (negative) temperature transport to the total OHT in the subtropical gyre, representing a return flow at depth and thus partially compensates the surface intensified temperature transport in $\text{HIST}_{mean}$ and $\text{RCP}_{mean}$.

In $\text{HIST}_{mean}$, the temperature transport of the NADW changes to northward (positive) transports in the subpolar gyre, signifi-



cantly increases north of 50°N and dominates the total OHT. Here, the NADW reaches the surface with outcropping isopycnals and thus includes both, the northward flow at the surface and the southward flow at depth and determines the total OHT in the northern SPG. In $\mathrm{RCP}_{mean}$ the temperature transport of the NADW is significantly reduced in the subpolar North Atlantic and yields southward temperature transports in the whole North Atlantic. This reflects, that the deep water formation in the North

Atlantic is significantly reduced and the isopycnals of the NADW do not outcrop anymore in the subpolar gyre. The temperature transport of the intermediate water shows only little changes, but replaces and even intensifies the northward temperature transport of the NADW in the subpolar gyre in $\mathrm{RCP}_{mean}$. The AABW shows only small transports in the North Atlantic in both $\mathrm{HIST}_{mean}$ and $\mathrm{RCP}_{mean}$.

## A2   Seasonal cycle in the temperature transport in potential density coordinates

When analyzing the seasonal cycle of the temperature transport in potential density coordinates we find a strong seasonal cycle in the temperature transport in the surface layer (Fig.12 c-d) in both $\mathrm{HIST}_{mean}$ and $\mathrm{RCP}_{mean}$ with seasonal amplitudes of about 3 PW and 2 PW, respectively. Between $\mathrm{HIST}_{mean}$ to $\mathrm{RCP}_{mean}$ the seasonal cycle pattern in the surface layer shifts significantly northward in the tropical and subtropical North Atlantic and thus alters the seasonal cycle between 20°N to 30°N with temporal shifts of 4 to 6 months in the minimum and maximum. Further, the seasonal cycle in the surface layer generally

intensifies in the subpolar gyre in $\mathrm{RCP}_{mean}$. The surface layer seasonal cycle can be assumed to be mostly wind-driven in the tropical North Atlantic and the subtropical gyre, so that the seasonal cycle also closely follows the Ekman heat transport seasonal cycle.

In the intermediate layer the temperature transport also indicates a relevant contribution to the OHT seasonal cycle (Fig.12 e-f). In the tropical and subtropical North Atlantic, the seasonal cycle of the intermediate water is mainly opposite to the seasonal

cycle of the surface layer in both $\mathrm{HIST}_{mean}$ and $\mathrm{RCP}_{mean}$ and thus partly compensates the seasonal cycle in the surface layer. From $\mathrm{HIST}_{mean}$ to $\mathrm{RCP}_{mean}$, the pattern shows shifts in the seasonal cycle of about 1 month to later months, but no clear northward shift as in the surface layer. The seasonal cycle in the subpolar gyre indicates general phase shift of up to 6 months from $\mathrm{HIST}_{mean}$ to $\mathrm{RCP}_{mean}$ with a shift of the maximum from summer to winter between approximately 40°N to 50°N and a shift of the maximum from winter to spring between 50°N and 60°N.

In the NADW (Fig.12 g-h), substantial changes occur resulting from changes in the water mass formation in the North Atlantic. In the $\mathrm{HIST}_{mean}$, the formation of NADW is present and leads to a seasonal cycle in the temperature transport of the NADW giving an important contribution especially in the subpolar gyre. In $\mathrm{RCP}_{mean}$ the seasonal cycle is weakened in the remaining temperature transport of the NADW with a decrease of the seasonal amplitude, thus showing a surface-ward shift of the processes acting on the OHT seasonal cycle especially in the subpolar gyre.

The AABW seasonal cycle is generally weak and thus does not significantly contribute to the full OHT seasonal cycle (Fig.12 i-j). Still, we find a seasonal cycle in $\mathrm{HIST}_{mean}$. In $\mathrm{RCP}_{mean}$ we find changes in the seasonal cycle with a northward shift of the pattern and also latitude dependent temporal shifts. These changes in the AABW might result from changed dynamics in the Southern Ocean also influencing the global ocean circulation, which we do not focus on in this study and thus need further analysis.



*Acknowledgements.* This work was supported by the Cooperative Project "RACE - Regional Atlantic Circulation and Global Change" funded through the German Federal Ministry for Education and Research (BMBF), 03F0651A (MF, JB), and by the Cluster of Excellence CliSAP (EXC177), Universität Hamburg, funded through the German Research Foundation (DFG) (DD and JB). The work of WM was supported by the German Federal Ministry for Education and Research (BMBF) project MiKlip (PT01LP1144A). Further, research leading to these

5   results has received funding from the European Union's Seventh Framework Programme (FP7/2007-2013) under grant agreement no. 308378 ENV.2012.6.1-1: Seasonal-to-decadal climate predictions towards climate services [http://www.specs-fp7.eu/]. The climate simulations were performed at the German Climate Computing Centre (DKRZ). We thank Ralf Hand for stimulating discussions.



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





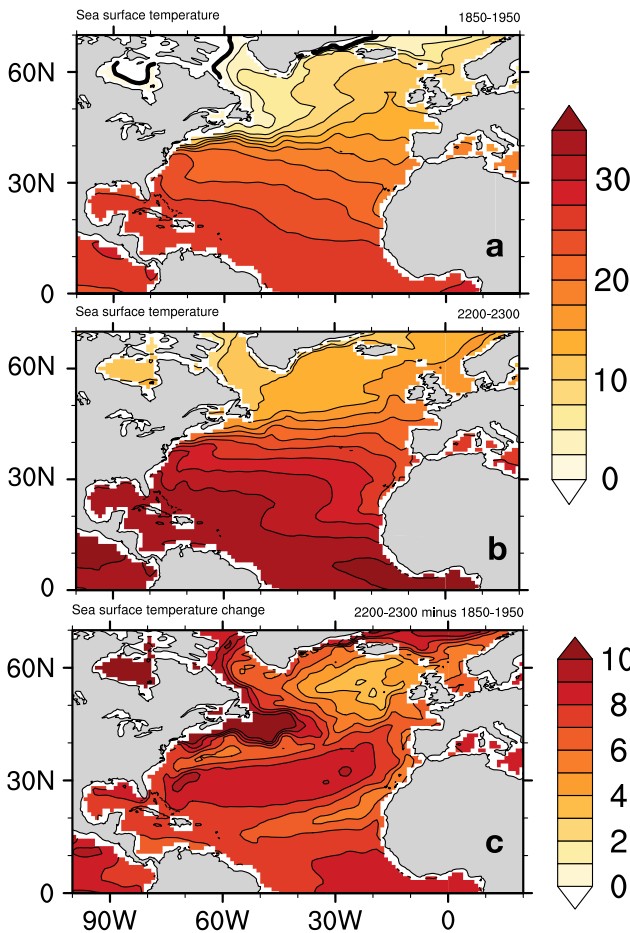

**Figure 1.** Sea surface temperature (°C) in **(a)** the historical simulation (1850-1950), **(b)** RCP8.5 (2200-2300) for the time mean and **(c)** difference between RCP8.5 and the historical simulation. Contour interval: $2.5°C$ in (a) and (b), $1°C$ in (c).




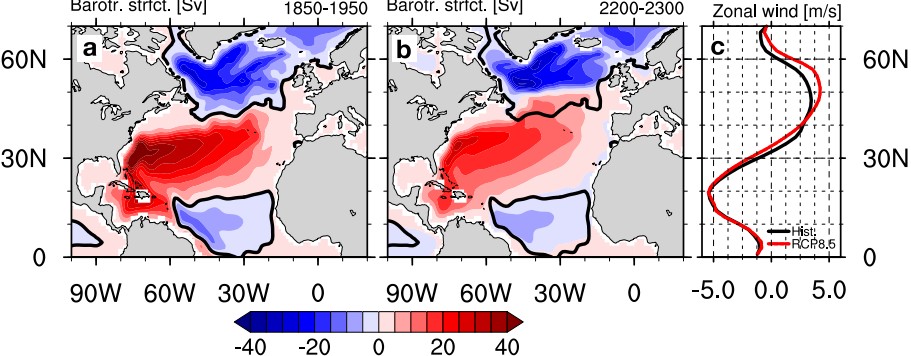

**Figure 2.** The barotropic streamfunction ($Sv = 10^6 m^3 s^{-1}$) in **(a)** the historical simulation (1850-1950) and **(b)** RCP8.5 (2200-2300) for the time mean over the respective period. The thick black line shows the zero contour in the historical simulation. Contour interval: 5 $Sv$. **(c)** Zonal mean zonal wind (at 1000 $hPa$) averaged over the North Atlantic region (90°W to 10°E ) for the historical simulation (black) and RCP8.5 (red) indicating the northward shift of the westerlies.

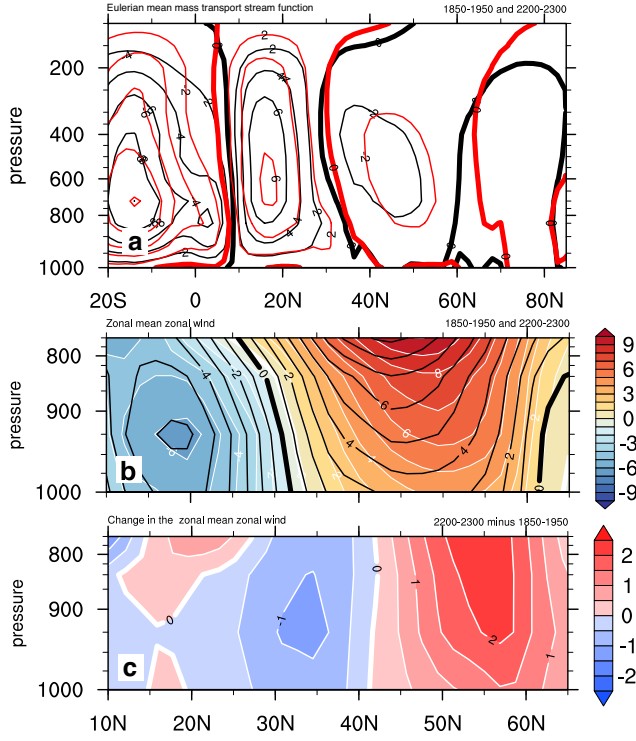

**Figure 3. (a)** The (global) Eulerian mean mass transport stream function (in $10^{10} kg/s$ with zonal averaging at fixed pressure for the historical simulation (1850-1950, black contours) and RCP8.5 (2200-2300, red contours) between 20°S and 80°N. **(b)** Vertical profile of the zonal-mean zonal wind ($ms^{-1}$) over the North Atlantic averaged from 10°E to 90°W in the historical simulation (1850-1950, black contours) and RCP8.5 (2200-2300, colors) for the time-mean over the respective periods. **(c)** The difference of the zonal-mean zonal wind between RCP8.5 and the historical simulation. Contour interval: $2 \times 10^{10} kg/s$ in (a), $1 m/s$ in (v), $0.5 m/s$ in (c).





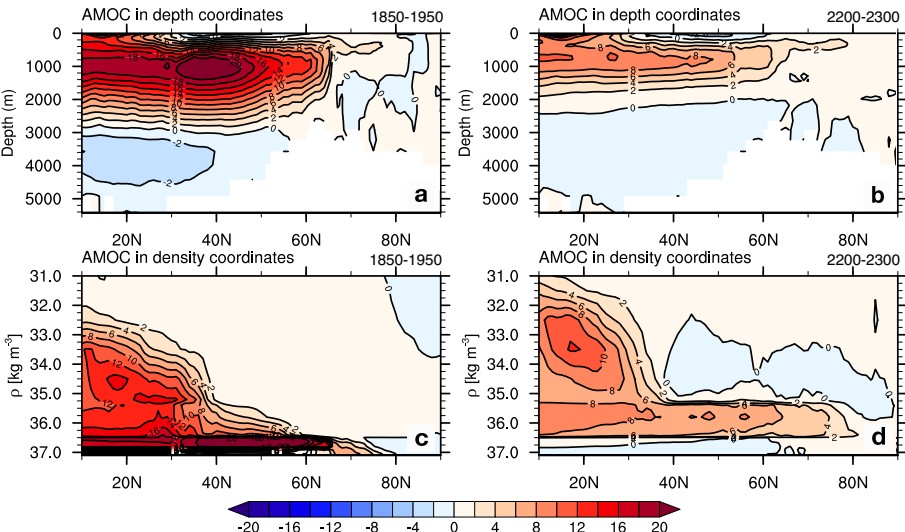

**Figure 4.** (a) and (b): AMOC in depth coordinates. (c) and (d): AMOC in density coordinates in the North Atlantic. (a) and (c) historical simulation (1850-1950), (b) and (d) RCP8.5 (2200-2300). Contour interval: 2 $Sv$.

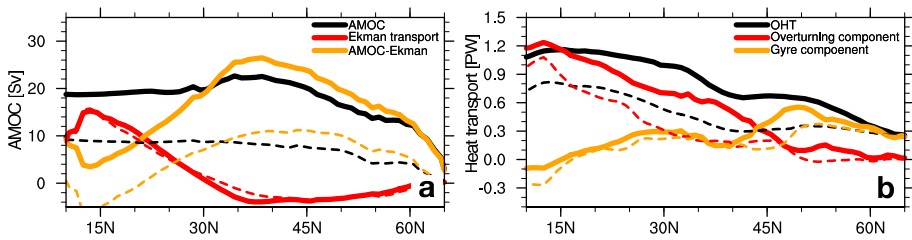

**Figure 5.** (a) Time-mean Atlantic meridional overturning circulation, the Ekman transport and the geostrophic volume transport ($\approx$AMOC-Ekman), (b) time-mean Atlantic meridional heat transport (OHT) with the overturning component and the gyre component (in $PW$). The historical simulation (1850-1950) is shown by solid lines, RCP8.5 (2200-2300) by dashed lines.





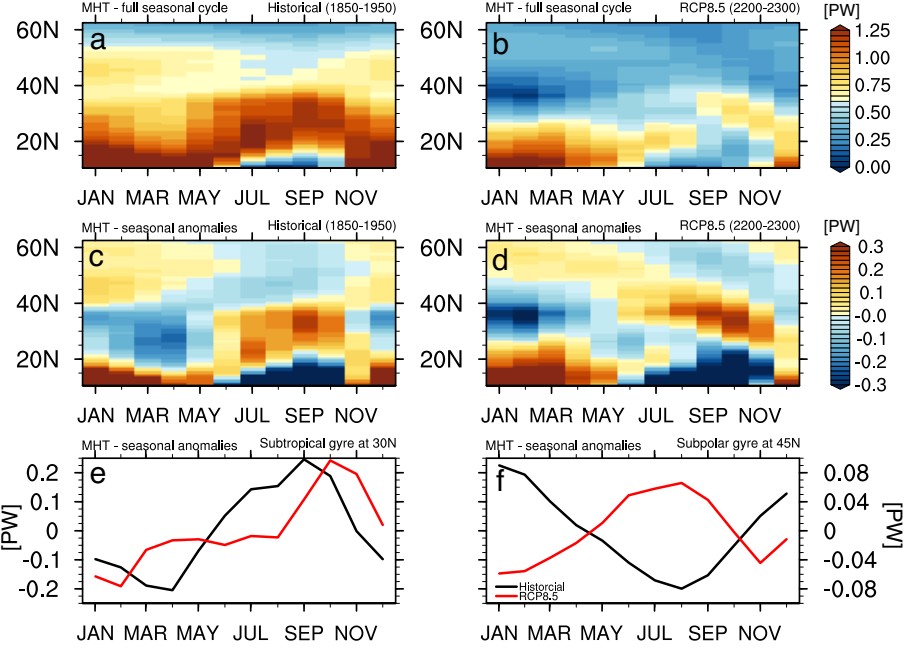

**Figure 6.** The Atlantic meridional heat transport seasonal cycle (in $PW$) in the historical simulation (1850-1950, **(a)** and **(c)**)) and RCP8.5 (2200-2300, **(b)** and **(d)**)). The OHT seasonal cycle in the historical simulation (1850-1950, black) and RCP8.5 (2200-2300, red) **(e)** at 30°N in the subtropical gyre and **(f)** at 45°N in the subpolar gyre. (a) and (b) show the full seasonal cyle, (c)-(f) show anomalies relative to the annual mean at every latitude. Colour interval in a-d: 0.02 $PW$.

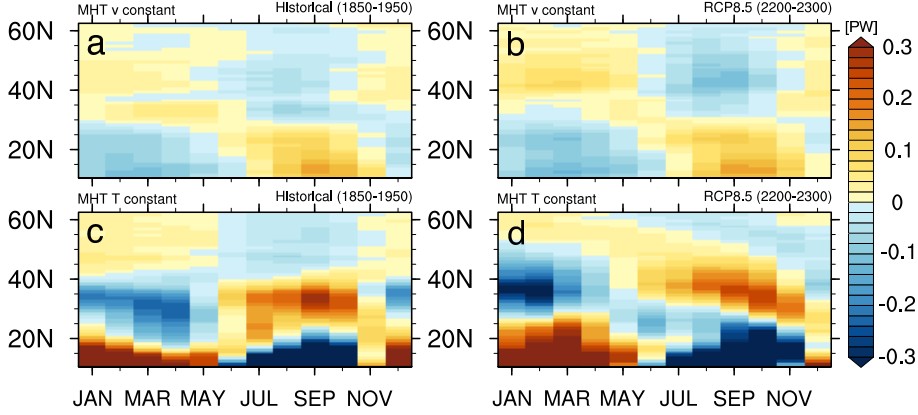

**Figure 7.** The Atlantic meridional heat transport seasonal cycle (in $PW$) in the historical simulation (1850-1950, **(a)** and **(c)**)) and RCP8.5 (2200-2300, **(b)** and **(d)**)) related to the variability in the temperature field (upper panels) and to variability in the velocity field (lower panels). Shown are anomalies relative to the annual mean at every latitude. Colour interval: 0.02 $PW$.





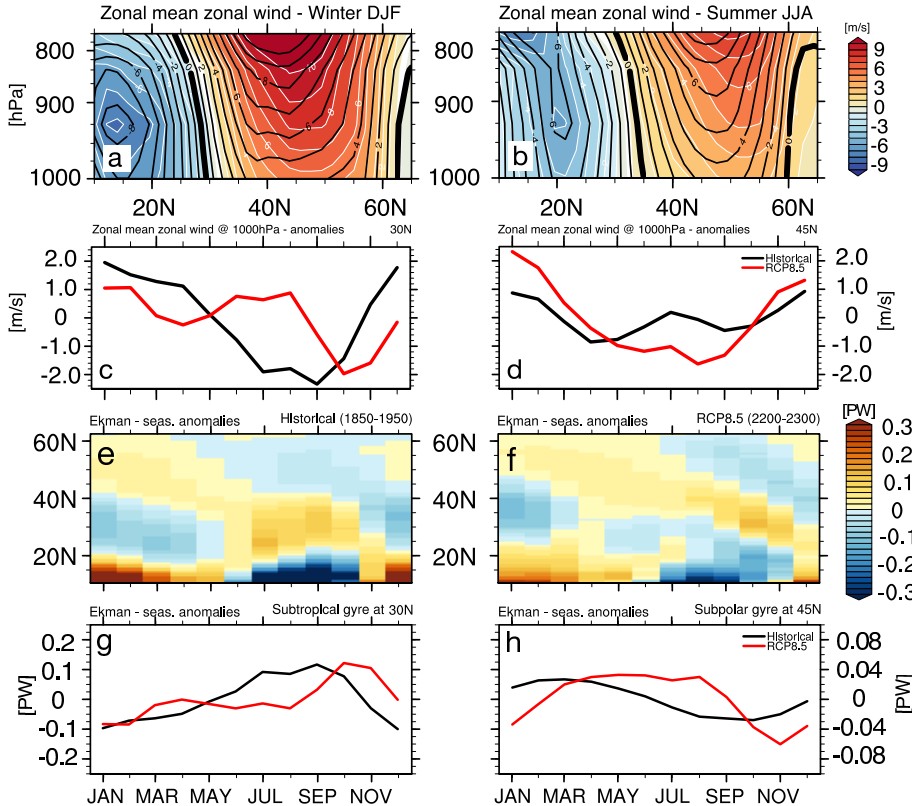

**Figure 8.** The zonal-mean zonal wind ($ms^{-1}$) over the North Atlantic averaged from $10°$E to $90°$W and the associated Ekman heat transport seasonal cycle ($PW$). **(a-b)** Vertical profile of the zonal wind for historical conditions (1850-1950, black contours) and RCP8.5 (2200-2300). Contour interval in **a** and **b**: 1 $m/s$. **(c-d)** Seasonal cycle of the surface wind at $30°$N and $45°$N for historical conditions (1850-1950, black) and RCP8.5 (2200-2300, red). **(e-f)**Seasonal cycle the Ekman heat transport (in $PW$) in the historical simulation (1850-1950, left panel) and RCP8.5 (2200-2300, right panel). **(g-h)** Seasonal cycle of the Ekman heat transport at $30°$N and $45°$N for historical conditions (1850-1950, black) and RCP8.5 (2200-2300, red). Shown are anomalies relative to the annual mean at every latitude. Contour interval in **e** and **f**: 0.02 $PW$.





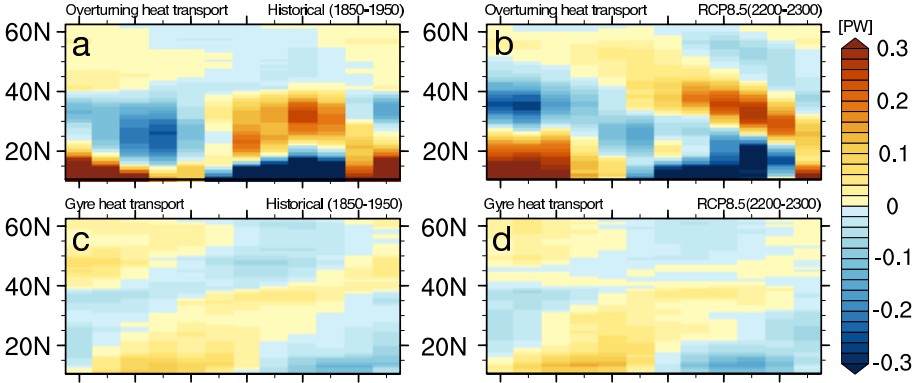

**Figure 9.** The seasonal cycle of **(a-b)** the overturning component and **(c-d)** the gyre component (in $PW$) in the historical simulation (1850-1950, left panel) and RCP8.5 (2200-2300, right panel). Shown are anomalies relative to the annual mean at every latitude. Contour interval: 0.02 $PW$.

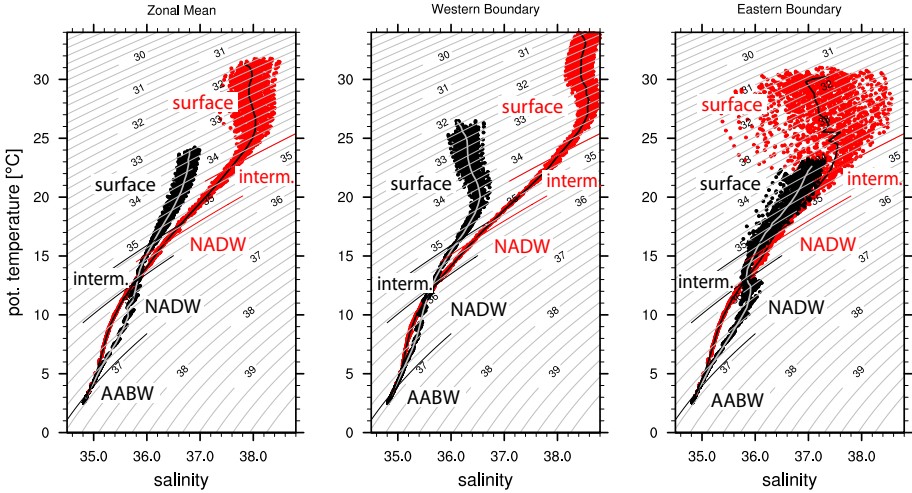

**Figure 10.** Temperature-salinity diagrams at 26°N for the historical simulation (1850-1950 in black) and RCP8.5 (2200-2300 in red) for **(a)** zonal mean, **(b)** western boundary and **(c)** eastern boundary temperatures and salinities. Water masses show the surface, intermediate, North Atlantic Deep Water (NADW) and Antarctic Bottom Water (AABW). The mean of the temperature-salinity diagrams averaged over density layers is shown in grey and black.





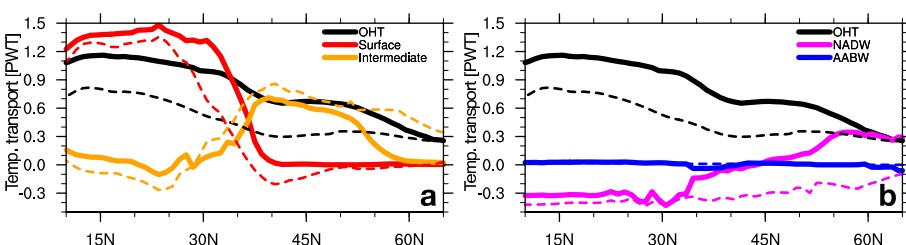

**Figure 11. (a)** Time-mean temperature transport in the surface layer (red) and intermediate layer (yellow, in $PWT$) compared to the total OHT (black) and **(b)** time-mean temperature transport in the North Atlantic Deep Water (NADW, magenta) and Antarctic Bottom Water (AABW, blue) (in $PWT$) compared to the total OHT. The historical simulation (1850-1950) is shown by solid lines, RCP8.5 (2200-2300) by dashed lines.





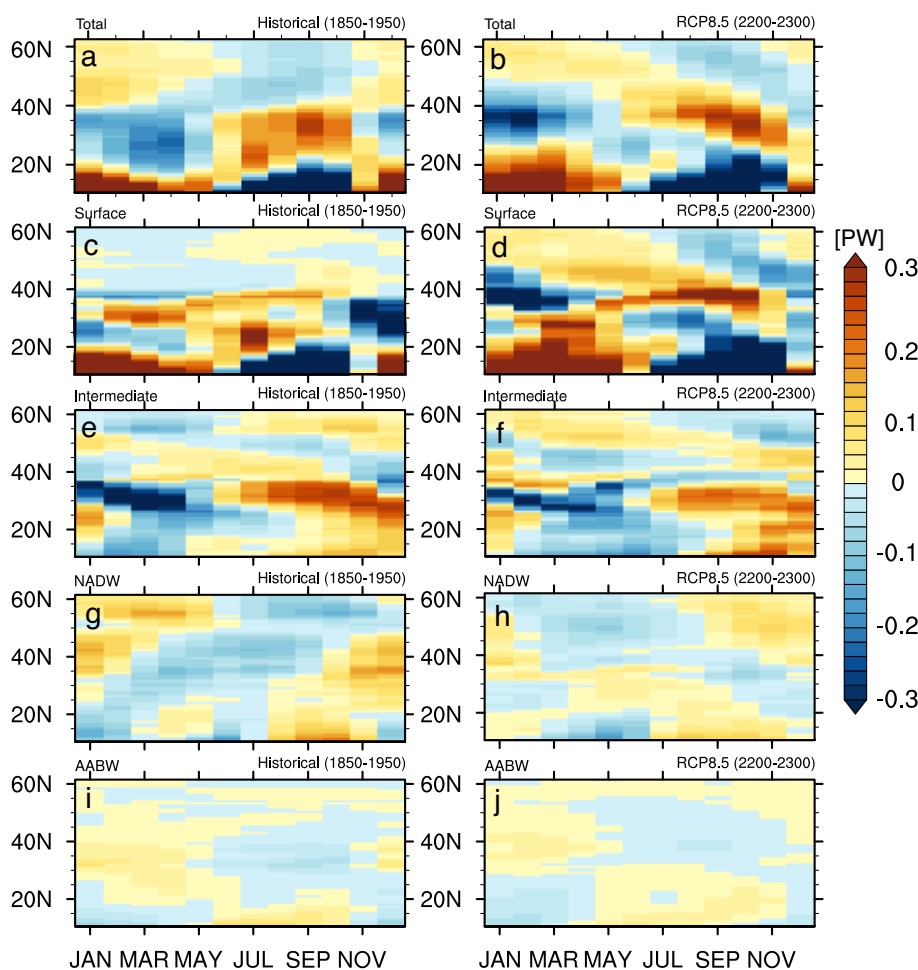

**Figure 12.** Contributions to the total OHT seasonal cycle from the temperature transport ($PWT$) of individual water masses calculated in potential density classes in the historical simulation (left) and RCP8.5 (right): **(a-b)** total OHT, **(c-d)** surface layer, **(e-f)** intermediate layer, **(g-h)** NADW and **(i-j)** AABW. Shown are anomalies relative to the annual mean at every latitude. Contour interval: 0.02 $PWT$.





**Table 1.** Definition of water masses

|  | surface | intermediate | NADW | AABW |
|---|---|---|---|---|
| $\text{HIST}_{mean}$ | $\sigma_2 \leq 35.2 kg/m^3$ | $35.2 kg/m^3 < \sigma_2 \leq 35.8 kg/m^3$ | $35.8 kg/m^3 < \sigma_2 \leq 36.91 kg/m^3$ | $\sigma_2 > 36.91 kg/m^3$ |
| $\text{RCP}_{mean}$ | $\sigma_2 \leq 34.5 kg/m^3$ | $34.5 kg/m^3 < \sigma_2 \leq 35.4 kg/m^3$ | $35.4 kg/m^3 < \sigma_2 \leq 36.91 kg/m^3$ | $\sigma_2 > 36.91 kg/m^3$ |