# Peer review of "Projected changes in the seasonal cycle of the Atlantic meridional heat transport in MPI-ESM"

_Earth System Dynamics, 2016_

## Referee Comment (RC1) · Anonymous Referee #1 · 20 Jul 2016

The authors examine the Atlantic meridional heat transport in their model in the present day and in the future and attempt attribute the cause of the changes. The paper is well written and clear. There is a lot of good background and the work is good. Technically, it fits into the scope of ESD as heat transport is very much a climate variable, not just an ocean variable, and it fits into aims and scope (2) Earth System Change. That said, the sister journal Ocean Science would also be a good fit.

I have two major issues that need addressing. Firstly, the shift in latitude of the seasonal cycle causes some shifts in time. This is not properly treated. I would suggest comparing the seasonal cycles of HISTmean and RCPmean shifted by five degrees to try to quantify this. See further comments below.

Secondly, One of the main conclusions of the paper is that all changes to the seasonal

[Figure]

cycle are explained by changes in the wind. Or at least that is the impression the reader gets. However, comparing Fig 8f with 6d although the wind explains most of what changes close to the equator, it does not have a magnitude large enough to explain what happens north of 30N. What is happening there? If there is a feedback that amplifies the signal or another contributing process this needs to be clearly written in the text, even if at this stage it is a little bit speculative.

Title: projected for when?

Page 1 Line 6: "For the total OHT seasonal cycle," I do not understand what follows this statement. The last part of the abstract needs to be revised so that it is clearer.

Page 1 Line 11: What changes in the gyre? Don't just say that it changes, say how it changes.

Page 2 Line 7: I don't think it is correct to say that the storm track will move northwards in light of the results of Zappa et al (2013).

Page 8 Line 11: "the equator the pole"?

Page 8 Line 19: "can not be fully explained by the northward shift" - which features do you refer to here? The following sentence indicates that it is at the gyre boundaries, but then the sentence after that claims the changes there "result from the northward shift", which leaves the reader confused. The only region that cannot to first order be explained as a northward shift is north of 50N, but as the values are so small there this may not be a robust result.

Page 9 Line 25: You should point out to the reader here that the seasonal cycle is more than three times larger in amplitude in the subtropics than in the subpolar gyre, just in case they do not look at the axis labels.

Page 10 Section 4.1.3: There are some references here to Fig 8, which I think are meant to be Fig 9. Otherwise this paper has no references to Figure 9!

Page 10 Lines 18-25: To my eye, the Figures show that the seasonal cycle of MHT between 30N-40N cannot be explained by the Ekman component. Figure 11 shows that NADW is changed at these latitudes, so could it be that this part is not wind driven but due entirely to the collapse of the AMOC? This is not acknowledged in this part of the text.

Page 11 Line 35: Although it is difficult to separate global warming and AMOC slow down in surface temperature, their footprints in outgoing longwave and absorbed short-wave radiation are very distinct, making attribution possible (Dirfjhout, 2015)

Page 12 Conclusion 1: Some of the shift in time is due to the shift in latitude. The way this conclusion is written it could be interpreted to mean that they are separate.

Page 12 Conclusion 4: Are the changes in the gyre heat transport seasonal cycle also due to wind-driven changes? It doesn't appear so from Figures 8 and 9. So what is causing it?

Fig 3 caption, last line: "(v)" should be "(b)"?

Fig 6 (e,f): you could add another line from RCPmean, which is the seasonal cycle 5 degrees further North. This would back up your statement on page 8 saying that the approximate shift of the pattern is 5 degrees. Though if these panels (and the equivalent ones in Fig 8) are meant to characterise the subtropical and subpolar gyre, then perhaps an average over a range of latitudes in each gyre would be better? After all, you wouldn't believe that the model can predict the climate change impact at one specific latitude, but you would be more confident that an average over most of the gyre is representative.

Fig 8 (a-b): What is the point of the vertical profile of the boundary layer only? It would be much more informative to have the winds at say 925hPa with latitude on the y-axis and month on the x-axis as in the other plots in this figure (which would be less confusing as well)

Sybren Difjhout (2015) Competition between global warming and an abrupt collapse of the AMOC in Earth's energy imbalance Scientific Reports 5, Article number: 14877 (2015)

Giuseppe Zappa, Len C. Shaffrey, Kevin I. Hodges, Phil G. Sansom, and David B. Stephenson (2013) A Multimodel Assessment of Future Projections of North Atlantic and European Extratropical Cyclones in the CMIP5 Climate Models Journal of Climate 2013 26:16, 5846-5862

---

## Referee Comment (RC2) · Anonymous Referee #2 · 27 Jul 2016

General Comment

In this manuscript the authors analyse how the seasonal cycle of the ocean heat transport in the Atlantic is affected by future climate change conditions, and the mechanisms responsible for these changes. The meridional ocean heat transport is known to be a key variable to understand the climate of the North Atlantic region. Thus, this analysis addresses convincingly a relevant scientific topic, by providing a mechanistic understanding of the potential future changes in the region.

Overall, I found the manuscript to be compelling and worthy of publication in Earth System Dynamics. The paper is well written and clear although there are some lingering points that need to be addressed.

I thus recommend acceptance pending a few revisions.

[Figure]

My major concern relates to the way that some of the results are presented. Many of the figures show equivalent panels for the historical and the RCP simulations. And these are often discussed in terms of the differences. However, I find that the changes usually discussed are not so evident when one looks at the plots. For example, the temporal shifts commented in lines 27-28 of page are hardly discernible in Fig 8e-f. As I see it, it would be more illustrative for the reader to present the figures differently. Instead of the separate patterns for the historical and the RCP simulations, it is more helpful to show one of the two (e.g. the panel of the historical run, which represents a baseline configuration) and then additionally a panel on the differences (historical-RCP), like in Fig. 3c. The main advantage is that this will show directly the actual changes that you discuss later on.

Another indirect benefit of showing the plots on the differences is that they allow including some statistical tests on the significance of the differences. These tests are actually key to identify which of the reported changes from the historical period to the climate change projections are actually significant, and which ones are probably due to climate noise. I strongly recommend the authors to include such tests on their plots.

Please, find a list of other specific comments below:

**1 [Page 1, lines 1-2]: As it is written, the authors seem to suggest that the changes in OHT's seasonal cycle appear in response to the overall OHT strength reduction. This is not exactly true. As I see it, both (the OHT strength weakening and the changes in its seasonal cycle) are simultaneously responding to the strong GHG forcing in the future projections.**

**2 [Page 2, line 1]: Please, substitute "expected" by "predicted".**

**3 [Page 2, line 15]: It could be one cause or another, or both causes. So I suggest changing "or" to "and/or".**

**4 [Page 2, line 34]: More than "to the ocean" in general they refer to "to internal ocean**

dynamics".

**5 [Page 3, line 10]: "Long-term variability" is too generic and depends on the length of the timeseries considered. The important thing to specify here is that they show decadal trends (which are an indicator of, at least, decadal variability in the overturning circulation and related OHT).**

**6 [Page 3, line 21]: I presume that you refer to the "meridional" overturning. Please, clarify in the text.**

**7 [Page 4, line 11]: Please, specify how this further increase is (Linear? Exponential?)**

**8 [Page 4, line 30; and other similar entries]: "zonal-mean zonal wind" is a bit confusing. I suggest "zonally-averaged zonal wind".**

**9 [Page 5, lines 2-4]: This sentence needs rephrasing. It is not to the NAO itself but to the zonal-wind pattern characteristic of a positive NAO that the shift in Fig 3b resembles. However, to support this claim, it would be good to include in Figure 3 an additional panel (Fig 3d?) showing simply the correlations between the NAO index and the zonally-averaged zonal winds. This result, to be confirmed, suggests also that the NAO is becoming more positive in the RCP runs. Have you checked if this is true?**

**10 [Page 5, lines 8-9]: You first say that there is "only a weak increase" in the gyre strength, and afterwards that this is "suggesting that changes in the deep circulation are important". Please, rephrase, as both things seem somehow contradictory.**

**11 [Page 6, lines 10-11]: Please, change to "The decomposition of . . . is well established by considering. . ." #12 #13 [Page 7, line 13]: Please, change "shifted to the surface" to "becomes s[]hallower" or "shoals". #14 [Page 7, line 16]: Please, rewrite as "The AMOC in density. . . indicates a similar shoaling of the AMOC cell. . ." #15 [Page 7, lines 17-18]: To guide the reader, I suggest to specify which are the levels involved in the wind-driven surface cell ($\sim$ upper 100m). Also, as opposed to this Ekman-driven cell, it would be good to mention that the deep cell mostly reflects the thermohaline**

circulation (as discussed in Kuhlbrodt et al 2007).

**16 [Page 8, line 11]: "from the Equator to the Pole".**

**17 [Page 8, line 15]: I suggest ending the sentence with "to thus highlight the seasonally varying changes."**

**18 [Page 8, lines 18-19]: It is not obvious to me how a northward shift can explain a temporal-shift.**

**19 [Page 9, line 19]: Remove "during summer" to avoid repetition (as it appears also in the same sentence in line18).**

**20 [Page 9, line 23 and Fig. 8g,h]: At first sight, the figure seems to suggest that the changes in the subpolar gyre are comparable to those in the subtropical gyre. Some readers might not notice that, indeed, the vertical axes are not the same in both panels. I suggest either to use the same scale in both cases, either to add something in the text like "please, notice that the vertical axes differ".**

**21 [Page 9, lines 32-33]: The sentence is confusing. Please, rephrase.**

**22 [Page 10, line 2]: Please, change to "similar than for".**

**23 [Page 10, line 3]: The first bracket for Fig. 8a,b is missing.**

**24 [Page 10, lines 3, 4, 9, 12]: I presume that you refer to Fig. 9 instead of Fig. 8.**

**25 [Page 10, line 10]: "determines changes" with respect to what?**

**26 [Page 10, line 13 and other similar entries]: Please avoid the use of "significant" as this adjective is commonly used for statistical analyses (which have not been considered here). I propose alternatives like "notable" or "remarkable".**

**27 [Page 10, line 25]: "Intermediate circulation" is not a term commonly used. I suggest upper mid-ocean circulation, or simply upper ocean circulation.**

**28 [Page 10, lines 30-32]: I don't follow. The two points made seem the same to me.**

[Figure]

Do you mean that the effect of the overturning dominates the intra-seasonal changes in the OHT, and also explains the differences in the OHT seasonal cycle from historical to RCP conditions? Please, clarify.

**29 [Page 10, line 33]: Please, change to "wind-driven via changes in the Ekman heat transport, which is mostly..."**

**30 [Page 11, line 2]: "as well as with changes"**

**31 [Page 11, line 10]: "remains under discussion"**

**32 [Page 11, line 12]: "show a poleward expansion"**

**33 [Page 11, line 18]: "and therefore in the associated"**

**34 [Page 12, line 5-6]: "Based on our analysis... we conclude for the Atlantic Ocean meridional heat transport that:"**

**35 [Page 12, line 22]: "vertical integral" of what?**

**36 [Page 13, line 1]: "with $\sigma 2$ being"**

Kuhlbrodt T, Griesel a, Montoya M, et al (2007) On the driving processes of the Atlantic meridional overturning circulation. Reviews of Geophysics 45: 2004RG000166.

---

## Author Comment (AC1) · 22 Nov 2016

Reply to reviewer #1

We thank the reviewer for carefully reading the manuscript and for the constructive comments. Below, we reply to all comments (starting with a (*)).

The authors examine the Atlantic meridional heat transport in their model in the present day and in the future and attempt attribute the cause of the changes. The paper is well written and clear. There is a lot of good background and the work is good. Technically, it fits into the scope of ESD as heat transport is very much a climate variable, not just an ocean variable, and it fits into aims and scope (2) Earth System Change. That said, the sister journal Ocean Science would also be a good fit.

[Figure]

(*) Thanks – indeed, we did ponder both journals, but decided for ESD to reach beyond the oceanographic community.

I have two major issues that need addressing. Firstly, the shift in latitude of the seasonal cycle causes some shifts in time. This is not properly treated. I would suggest comparing the seasonal cycles of HISTmean and RCPmean shifted by five degrees to try to quantify this. See further comments below.

(*) Thanks. The figure is an interesting suggestion, yet, the shift in latitude is not so uniform that with its help we could quantify the shift in the seasonal cycle further.

Secondly, One of the main conclusions of the paper is that all changes to the seasonal cycle are explained by changes in the wind. Or at least that is the impression the reader gets. However, comparing Fig 8f with 6d although the wind explains most of what changes close to the equator, it does not have a magnitude large enough to explain what happens north of 30N. What is happening there? If there is a feedback that amplifies the signal or another contributing process this needs to be clearly written in the text, even if at this stage it is a little bit speculative.

(*) We carefully checked whether this notion appears in the abstract and added the following statement to the conclusions: "In the subpolar North Atlantic, we also find that the reduction of the North Atlantic Deep Water formation result in a weakened seasonal cycle with a weakened seasonal amplitude by the end of the 23rd century and thus changes the OHT seasonal cycle in the SPG." Also, see below, in response to your further comments, for additional places, where we changed this notion.

Title: projected for when?

(*) We changed the title to: 'Changes in the seasonal cycle of the Atlantic meridional heat transport in a RCP 8.5 climate projection in MPI-ESM'

Page 1 Line 6: "For the total OHT seasonal cycle," I do not understand what follows this statement. The last part of the abstract needs to be revised so that it is clearer.

(*) We revised the abstract and now mention the reference period: "We investigate changes in the seasonal cycle of the Atlantic Ocean meridional heat transport (OHT) in a climate projection experiment with the Max-Planck Institute Earth System Model (MPI-ESM) performed for the Coupled Model Intercomparison Project phase 5 (CMIP5). Specifically, we compare a RCP8.5 climate change scenario, covering the simulation period from 2005 to 2300, against a historical simulation, covering the simulation period from 1850 to 2005. In RCP8.5, the OHT declines in comparison to the historical simulation in the North Atlantic by 30-50\% by the end of the 23rd century. The decline in the OHT is accompanied by a change in the seasonal cycle of the total OHT and its components. We decompose the OHT into overturning and gyre component. For the OHT seasonal cycle, we find a northward shift of 5 degrees and latitude-dependent shifts between 1 and 6 months that are mainly associated with changes in the meridional velocity field. We find that the changes in the OHT seasonal cycle predominantly result from changes in the wind-driven surface circulation which projects onto the overturning component of the OHT in the tropical and subtropical North Atlantic. This leads in turn to latitude-dependent shifts between 1 and 6 months in the overturning component. In the subpolar North Atlantic, we find in RCP8.5 in comparison to the historical simulation, a reduction of the North Atlantic Deep Water formation and changes in the gyre heat transport result in a strongly weakened seasonal cycle with a weakened amplitude by the end of the 23rd century."

Page 1 Line 11: What changes in the gyre? Don't just say that it changes, say how it changes.

(*) We omit this statement in the abstract, since we could – within the constraints – not elaborate enough on the changes in the subpolar gyre (in addition to what is mentioned for the gyre heat transport before).

Page 2 Line 7: I don't think it is correct to say that the storm track will move northwards in light of the results of Zappa et al (2013).

(*) We agree that the storm track response is more complex than a simple northward shift. We have therefore changed this sentence to read: "The atmospheric circulation patterns are projected to move poleward in concert with the warming of surface temperature, leading to a poleward expansion of the tropical cell and an associated poleward shift of the jet stream (Chang et al., 2012; Hu et al., 2013; IPCC, 2013), while the response of the storm track exhibits a more complex pattern (Zappa et al., 2013)."

Reference: Zappa, G., Shaffrey, L. C., Hodges, K. I., Sansom, P. G., Stephenson, D. B., Zappa, G., et al. (2013). A Multimodel Assessment of Future Projections of North Atlantic and European Extratropical Cyclones in the CMIP5 Climate Models*. Dx.Doi.org. http://doi.org/10.1175/JCLI-D-12-00573.1

Page 8 Line 11: "the equator the pole"?

(*) Corrected.

Page 8 Line 19: "can not be fully explained by the northward shift" - which features do you refer to here? The following sentence indicates that it is at the gyre boundaries, but then the sentence after that claims the changes there "result from the northward shift", which leaves the reader confused. The only region that cannot to first order be explained as a northward shift is north of 50N, but as the values are so small there this may not be a robust result.

(*) Reworded to explain that the seasonal cycle at a given place could be changed as a result from an overall northward shift, but this does not fully explain the entire shift.

Page 9 Line 25: You should point out to the reader here that the seasonal cycle is more than three times larger in amplitude in the subtropics than in the subpolar gyre, just in case they do not look at the axis labels.

(*) We included the following sentence in the figure caption: Please note the different vertical axes in c,d and g,h.

Page 10 Section 4.1.3: There are some references here to Fig 8, which I think are

meant to be Fig 9. Otherwise this paper has no references to Figure 9!

(*) Thanks. Corrected to figure 9.

Page 10 Lines 18-25: To my eye, the Figures show that the seasonal cycle of MHT between 30N-40N cannot be explained by the Ekman component. Figure 11 shows that NADW is changed at these latitudes, so could it be that this part is not wind driven but due entirely to the collapse of the AMOC? This is not acknowledged in this part of the text.

(*) We included a sentence: "Note that between 30N and 40N, the Ekman transport change alone cannot explain the changes in the seasonal cycle of the OHT, though these latitudes are strongly influenced by changes in the mean strength of the North Atlantic Deep Water (appendix A)."

Page 11 Line 35: Although it is difficult to separate global warming and AMOC slow down in surface temperature, their footprints in outgoing longwave and absorbed short-wave radiation are very distinct, making attribution possible (Dirfjhout, 2015)

(*) We included a sentense to cite this study: "This yields an overall increase in surface temperature in the North Atlantic sector, which maybe be possible to separate from an AMOC decrease due to their distinctive footprints in outgoing longwave and absorbed shortwave radiation (Drijfhout, 2015)."

Page 12 Conclusion 1: Some of the shift in time is due to the shift in latitude. The way this conclusion is written it could be interpreted to mean that they are separate.

(*) Changed to read: "Accompanied by a 30 to 50% decline of the time-mean OHT, the seasonal cycle of the OHT shifts in time (1 to 6 months, depending on latitude and season) and in space (5 degrees northward) in both the subtropical and subpolar gyre in RCP8.5."

Page 12 Conclusion 4: Are the changes in the gyre heat transport seasonal cycle also due to wind-driven changes? It doesn't appear so from Figures 8 and 9. So what is

causing it?

(*) Changes in the gyre component might result from changes in the wind as well, but are likely to be the results of zonally-varying changes in the wild variability. We reworded the conclusion: "Thus, the changes in the total OHT seasonal cycle in the subtropical gyre result mostly from the zonal-mean wind-driven and surface-intensified part of the overturning heat transport, whereas in the subpolar gyre, the changes in the seasonal cycle are dominated by the gyre heat transport."

Fig 3 caption, last line: "(v)" should be "(b)"?

(*) Corrected.

Fig 6 (e,f): you could add another line from RCPmean, which is the seasonal cycle 5 degrees further North. This would back up your statement on page 8 saying that the approximate shift of the pattern is 5 degrees. Though if these panels (and the equivalent ones in Fig 8) are meant to characterise the subtropical and subpolar gyre, then perhaps an average over a range of latitudes in each gyre would be better? After all, you wouldn't believe that the model can predict the climate change impact at one specific latitude, but you would be more confident that an average over most of the gyre is representative.

(*) Indeed, we would like to keep the individual lines as a mere illustration for what the seasonal cycle at an individual latitude looks like, since the actual changes are not only latitude dependent but their structure may be very model dependent. For the conclusions, we would therefore like the reader to focus on the overall picture, which is why we did not add another line into the subpanels.

Fig 8 (a-b): What is the point of the vertical profile of the boundary layer only? It would be much more informative to have the winds at say 925hPa with latitude on the y-axis and month on the x-axis as in the other plots in this figure (which would be less confusing as well)

(*) We included this figure with the vertical axis to be comparable to figure 3. We think it is important to be able to relate the discussion of figure 8 to the discussion of figure 3, and therefore use the same setup. The information of latitude vs months can be drawn from the other available figures.

―――――――――――――――――

---

## Author Comment (AC2) · 22 Nov 2016

Reply to reviewer #2

We thank the reviewer for carefully reading the manuscript and the constructive comments. Below, we reply to all comments (starting with a (*)).

General Comment In this manuscript the authors analyse how the seasonal cycle of the ocean heat transport in the Atlantic is affected by future climate change conditions, and the mechanisms responsible for these changes. The meridional ocean heat transport is known to be a key variable to understand the climate of the North Atlantic region. Thus, this analysis addresses convincingly a relevant scientific topic, by providing a mechanistic understanding of the potential future changes in the region.

[Figure]

Overall, I found the manuscript to be compelling and worthy of publication in Earth System Dynamics. The paper is well written and clear although there are some lingering points that need to be addressed.

I thus recommend acceptance pending a few revisions. Printer-friendly version My major concern relates to the way that some of the results are presented. Many of the figures show equivalent panels for the historical and the RCP simulations. And these are often discussed in terms of the differences. However, I find that the changes usually discussed are not so evident when one looks at the plots. For example, the temporal shifts commented in lines 27-28 of page are hardly discernible in Fig 8e-f. As I see it, it would be more illustrative for the reader to present the figures differently. Instead of the separate patterns for the historical and the RCP simulations, it is more helpful to show one of the two (e.g. the panel of the historical run, which represents a baseline configuration) and then additionally a panel on the differences (historical-RCP), like in Fig. 3c. The main advantage is that this will show directly the actual changes that you discuss later on.

(*) Thank you for this thoughtful comment. Prior to submission of the manuscript, we did test various ways of illustrating the results. The difference plots are an obvious candidate. Yet, for shifts in the seasonal cycle that occur on one or both axes, the differences plots are unfortunately not as clear as one would hope. After careful consideration (at that time and now again after your comment), we still decided to show the fields; which also makes clear that sometimes the differences happen to be small.

Another indirect benefit of showing the plots on the differences is that they allow including some statistical tests on the significance of the differences. These tests are actually key to identify which of the reported changes from the historical period to the climate change projections are actually significant, and which ones are probably due to climate noise. I strongly recommend the authors to include such tests on their plots.

(*) We agree. Yet, given the length of time series, and the focus on the physics, we

decided to leave out a statistical analysis, whose assumptions would in the given case influence the result considerably. Following your comment, we did change the notion in the entire manuscript, focusing on the changes and their possible physical reasons.

Please, find a list of other specific comments below:

**1 [Page 1, lines 1-2]: As it is written, the authors seem to suggest that the changes in OHT's seasonal cycle appear in response to the overall OHT strength reduction. This is not exactly true. As I see it, both (the OHT strength weakening and the changes in its seasonal cycle) are simultaneously responding to the strong GHG forcing in the future projections.**

(*) Thanks. We reworded the sentence to read: "We investigate changes in the seasonal cycle of the Atlantic Ocean meridional heat transport (OHT) in a climate projection experiment with the Max-Planck Institute Earth System Model (MPI-ESM) performed for the Coupled Model Intercomparison Project phase 5 (CMIP5)."

**2 [Page 2, line 1]: Please, substitute "expected" by "predicted".**

(*) Replaced by 'projected', since we anticipate to build on the present study with a multi-year prediction study.

**3 [Page 2, line 15]: It could be one cause or another, or both causes. So I suggest changing "or" to "and/or".**

(*) Changed as suggested.

**4 [Page 2, line 34]: More than "to the ocean" in general they refer to "to internal ocean dynamics".**

(*) Changed as suggested.

**5 [Page 3, line 10]: "Long-term variability" is too generic and depends on the length of the timeseries considered. The important thing to specify here is that they show decadal trends (which are an indicator of, at least, decadal variability in the overturning**

circulation and related OHT).

(*) Replaced 'long-term' with 'interannual', as the time series are just over a decade long.

**6 [Page 3, line 21]: I presume that you refer to the "meridional" overturning. Please, clarify in the text.**

(*) Changed to 'meridional overturning circulation'.

**7 [Page 4, line 11]: Please, specify how this further increase is (Linear? Exponential?)**

(*) Corrected to 'stabilized'.

**8 [Page 4, line 30; and other similar entries]: "zonal-mean zonal wind" is a bit confusing. I suggest "zonally-averaged zonal wind".**

(*) While maybe sounding a bit cumbersome at first, "zonal-mean zonal wind" is a commonly used term in atmospheric dynamics. Examples are e.g.Âă

Barriopedro, D., & Calvo, N. (2014). On the Relationship between ENSO, Stratospheric Sudden Warmings, and Blocking. Journal of Climate, 27(12), 4704–4720. http://doi.org/10.1175/JCLI-D-13-00770.1

Birner, T., & Williams, P. D. (2008). Sudden stratospheric warmings as noise-induced transitions. Journal of the Atmospheric Sciences, 65(10), 3337–3343. http://doi.org/10.1175/2008JAS2770.1

**9 [Page 5, lines 2-4]: This sentence needs rephrasing. It is not to the NAO itself but to the zonal-wind pattern characteristic of a positive NAO that the shift in Fig 3b resembles. However, to support this claim, it would be good to include in Figure 3 an additional panel (Fig 3d?) showing simply the correlations between the NAO index and the zonally-averaged zonal winds. This result, to be confirmed, suggests also that the NAO is becoming more positive in the RCP runs. Have you checked if this is true?**

(*) Yes, the NAO is becoming more positive in the RCP simulation. However, correlating the changes in the wind pattern to the NAO is rather complicated due to the changes in the NAO pattern itself between pre-industrial control and RCP scenarios. Ning & Bradley (2016) find that the centers of the NAO loading patterns change considerably in the strong RCP scenarios, and the NAO pattern to project onto it therefore aÂă"moving target". We have therefore rephrased the sentence to include this notion:Âă"As a consequence, the westerlies between 30N and 60N are shifted poleward in RCP8.5 by about 5 degrees (Fig.3b,c). This shift resembles the wind pattern observed during a positive NAO anomaly (as defined from pre-industrial control, while the loading pattern may change considerably with climate change (Ning & Bradley, 2016)), which is associated with an acceleration of the westerlies over large areas of the SPG (Fig.3b,c), along with a deceleration of the westerlies between 30N - 40N and a slight intensification of the trade winds south of 30N."Âă

Reference: Ning, L., & Bradley, R. S. (2016). NAO and PNA influences on winter temperature and precipitation over the eastern United States in CMIP5 GCMs.ÂăClimate Dynamics,Âă46(3-4), 1257–1276. http://doi.org/10.1007/s00382-015-2643-9

**10 [Page 5, lines 8-9]: You first say that there is "only a weak increase" in the gyre strength, and afterwards that this is "suggesting that changes in the deep circulation are important". Please, rephrase, as both things seem somehow contradictory.**

(*) Reworded to bring out clearer that the flat bottom Sverdrup transport is only weakly increasing, hence not explaining the entire increase: "In particular, the flat-bottom Sverdrup transport in the subpolar gyre indicates only a weak increase of about 0.5 Sv in the gyre strength from HISTmean to RCPmean (not shown), suggesting that changes in the deep circulation might also be important (Greatchbach 1991)."

**11 [Page 6, lines 10-11]: Please, change to "The decomposition of . . . is well established by considering. . ."**

(*) Changed as suggested.

**12**

(*) #12 was missing in the reviewer comment. Please let us know in case this was more than a formatting problem.

**13 [Page 7, line 13]: Please, change "shifted to the surface" to "becomes shallower" or "shoals".**

(*) Changed to "is reduced in strength and becomes shallower".

**14 [Page 7, line 16]: Please, rewrite as "The AMOC in density... indicates a similar shoaling of the AMOC cell..."**

(*) Changed as suggested.

**15 [Page 7, lines 17-18]: To guide the reader, I suggest to specify which are the levels involved in the wind-driven surface cell ($\sim$ upper 100m). Also, as opposed to this Ekman-driven cell, it would be good to mention that the deep cell mostly reflects the thermohaline circulation (as discussed in Kuhlbrodt et al 2007).**

(*) Changed as suggested.

**16 [Page 8, line 11]: "from the Equator to the Pole".**

(*) Corrected.

**17 [Page 8, line 15]: I suggest ending the sentence with "to thus highlight the seasonally varying changes."**

(*) Thanks. Changed as suggested.

**18 [Page 8, lines 18-19]: It is not obvious to me how a northward shift can explain a temporal-shift.**

(*) Reworded to explain that the seasonal cycle at a given place could be changed as a result from an overall northward shift, but this does not fully explain the entire shift.

[Figure]

**19 [Page 9, line 19]: Remove "during summer" to avoid repetition (as it appears also in the same sentence in line18).**

(*) Removed as suggested.

**20 [Page 9, line 23 and Fig. 8g,h]: At first sight, the figure seems to suggest that the changes in the subpolar gyre are comparable to those in the subtropical gyre. Some readers might not notice that, indeed, the vertical axes are not the same in both panels. I suggest either to use the same scale in both cases, either to add something in the text like "please, notice that the vertical axes differ".**

(*) We included the following sentence in the figure caption: Please note the different vertical axes in c,d and g,h.

**21 [Page 9, lines 32-33]: The sentence is confusing. Please, rephrase.**

(*) Reworded to: "Overall, the seasonal cycle of the Ekman heat transport changes depending on latitude, closely following the changes in the seasonal cycle of the surface wind."

**22 [Page 10, line 2]: Please, change to "similar than for".**

(*) Corrected.

**23 [Page 10, line 3]: The first bracket for Fig. 8a,b is missing.**

(*) Corrected.

**24 [Page 10, lines 3, 4, 9, 12]: I presume that you refer to Fig. 9 instead of Fig. 8.**

(*) Thanks. Corrected.

**25 [Page 10, line 10]: "determines changes" with respect to what?**

(*) Reworded to reflect that changes in the amplitude of the seasonal cycle of the overturning component result in changes in the amplitude of the seasonal cycle of the total OHT.

**26 [Page 10, line 13 and other similar entries]: Please avoid the use of "significant" as this adjective is commonly used for statistical analyses (which have not been considered here). I propose alternatives like "notable" or "remarkable".**

(*) Replaced here and elsewhere.

**27 [Page 10, line 25]: "Intermediate circulation" is not a term commonly used. I suggest upper mid-ocean circulation, or simply upper ocean circulation.**

(*) Changed to "upper ocean".

**28 [Page 10, lines 30-32]: I don't follow. The two points made seem the same to me. Do you mean that the effect of the overturning dominates the intra-seasonal changes in the OHT, and also explains the differences in the OHT seasonal cycle from historical to RCP conditions? Please, clarify.**

(*) We removed the first part of the sentence.

**29 [Page 10, line 33]: Please, change to "wind-driven via changes in the Ekman heat transport, which is mostly..."**

(*) Changes as suggested.

**30 [Page 11, line 2]: "as well as with changes"**

(*) Corrected.

**31 [Page 11, line 10]: "remains under discussion"**

(*) Corrected.

**32 [Page 11, line 12]: "show a poleward expansion"**

(*) Changed as suggested.

**33 [Page 11, line 18]: "and therefore in the associated"**

(*) Changed as suggested.

**34 [Page 12, line 5-6]: "Based on our analysis... we conclude for the Atlantic Ocean meridional heat transport that:"**

(*) Changed as suggested.

**35 [Page 12, line 22]: "vertical integral" of what?**

(*) Added: "...of the temperature and meridional velocity fields"

**36 [Page 13, line 1]: "with _2 being"**

(*) Corrected.